# RECONSTRUCTING EVOLUTIONARY TRAJECTORIES OF MUTATIONS IN CANCER

**Yulia Rubanova, Ruian Shi, Roujia Li, Jeff Wintersinger,**
**Amit Deshwar, Nil Sahin, PCAWG-11 working group, Quaid Morris**
University of Toronto
`quaid.morris@utoronto.ca`

## ABSTRACT

We present a new method, TrackSig, to estimate evolutionary trajectories in cancer. Our method represents cancer evolution in terms of mutational signatures – multinomial distributions over mutation types. TrackSig infers an approximate order in which mutations accumulated in cancer genome, and then fits the signatures to the mutation time series. We assess TrackSig's reconstruction accuracy using simulations. We find 1.9% median discrepancy between estimated mixtures and ground truth. The size of the signature change is consistent in 87% cases and direction of change is consistent in 95% of cases. The code is available at `https://github.com/YuliaRubanova/TrackSig`.

## 1 INTRODUCTION

Mutations accumulate throughout our lifetime. Cancerous cells accumulate 100s to 1000s of mutations due to different causes that can be external (e.g., smoking, exposure to UV light) or internal (e.g. copy errors, failure of DNA damage repair). Different mutation sources create unique distributions over types of new mutations, known as a mutational signatures. A *mutational signature* is a probability distribution over these 96 types, where each element is a probability of generating mutations from the corresponding type Alexandrov et al. (2013b). Each signature is assigned an *exposure* which represents the proportion of mutations that the signature generates. In machine learning terms, mutation signatures correspond to topics in a topic model over vocabulary of mutation types, and mixing coefficients represent exposures. Many of the signatures are strongly associated with known mutational processes including smoking Alexandrov et al. (2016), DNA repair Alexandrov et al. (2013a) and ionizing radiation Behjati et al. (2016). Signature analysis can predict cancer prognosis McPherson et al. (2016) or guide treatment choice Davies et al. (2017).

Mutational sources can change over time Gerstung et al. (2017); Yates et al. (2017). Recent analyses use mutation clustering to investigate signature changes over time. However, the accuracy of these methods relies heavily on clustering methods, which typically have low precision Deshwar et al. (2015). We have developed a variant of dynamic topic modelling to reconstruct signature exposures across linear time, without using any clustering methods.

## 2 METHODS

TrackSig has two stages. First, we approximately order mutations in time. Next, we infer a trajectory of the mutational signature exposures over the inferred mutation ordering. We fit the exposure for each signature is a piecewise constant function with a small number of changepoints.

### 2.1 ESTIMATING MUTATION ORDER

To estimate the order of acquisition of mutations, we sort mutations according to decreasing variant allele frequency and use it as an estimate the order in which mutations were acquired. Given this ordering, we partition the sorted set of mutations into bins of hundred mutations and interpret each bin as one time point. Note that we are only using mutations from a single tumor sample.

## 2.2 Computing exposure to mutational signatures

We apply topic modeling Blei (2012) to infer signature exposures.Withing the time point, we separate mutation separate into K mutation types. Mutation types relate to vocabulary in topic modeling. A mutation process is represented as a distribution over this vocabulary, known as a "*mutation signature*". We calculate the *exposure* of a set of mutations to different processes by estimating the proportion of its mutations generated by each process.

We represent each mutation as one-hot-encoding of a mutation type. We will denote $\mathbf{x}^{(n)}$ to be the one-hot-encoding of mutation $n$. A sample containing $N$ mutations is represented as a $N \times K$ binary matrix $\mathbf{X}$, where each column corresponds one mutation. A mutation process is represented as a distribution over mutation types, known as a "*mutation signature*". We will denote signature multinomials as $K$-dimensional probability vectors $\boldsymbol{\mu}_i$, where $i = \{1..M\}$ is an index over signatures. Signatures are fixed and are not updated during the training. We aim to estimate signature *exposures* $\boldsymbol{\pi}$ – the proportion of mutations generated by each signature.

We represent mutation matrix $\mathbf{X}$ as a mixture of signature multinomials $\boldsymbol{\mu_1}, ..\boldsymbol{\mu_K}$ with mixture coefficients $\boldsymbol{\pi}$:

$$\mathbf{X} \sim \text{Multinomial}(N; \textstyle\sum_{i=1}^{M} \pi_i \, \boldsymbol{\mu_i})$$

We denote $z_n$ to be the signature assignment of mutation $n$. The probability of a mutation $n$ to be generated by signature $i$ is given by:

$$p(\mathbf{x}^{(n)}|z_n = i, \boldsymbol{\pi}, \boldsymbol{\mu_1}, ..\boldsymbol{\mu_K}) = \textstyle\prod_{k=1}^{K} \mu_{ik}^{x_k^{(n)}}; \quad i \in \{1..M\}; \quad n \in \{1..N\}$$

Log likelihood of the collection of mutations in a sample:

$$\log \mathcal{L}(\mathbf{X}|\boldsymbol{\pi}, \boldsymbol{\mu_1}, ..\boldsymbol{\mu_K}) = \textstyle\sum_{n=1}^{N} \log \sum_{i=1}^{M} p(\mathbf{x}^{(n)}|z_n = i, \boldsymbol{\pi}, \boldsymbol{\mu_1}, ..\boldsymbol{\mu_K}) p(z_n = i|\boldsymbol{\pi})$$

We fit mixing coefficients $\boldsymbol{\pi}$ in each bin using Expectation-Maximization (EM) Dempster et al. (1977). EM starts with uniform mixing coefficients and iterates between updating posterior distribution over $z_n$ and updating an estimate of the mixing coefficients $\boldsymbol{\pi}$. Resulting mixture coefficients are the exposures of the mutational signatures, i.e. proportion of mutations generated by a certain cell process. Mixture coefficients at each time point form trajectories that span over time and reflect how the exposure to cell processes (i.e. signatures) was changing during tumor development.

## 2.3 Detecting change-points

We are interested to find the time points, when signature profiles change substantially (change points), as these represent loss or gain of different mutational processes. To detect the signature change points, we iterate over the time points and recompute the multinomial mixtures separately before and after change point. Thus, exposure trajectories obtain piecewise constant shape. We estimate the likelihood of the data given the potential change point. A point with maximum likelihood is considered a new change point. We use Bayesian Information Criterion (BIC) to determine the optimal number of change points.

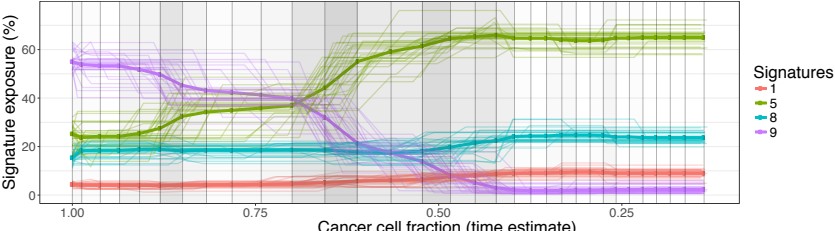

Figure 1: **An example of trajectories in one leukemia sample.** Each line is an exposure trajectory for a single signature (y-axis) as a function of time (x-axis). Thin lines are trajectories from each of 30 bootstrap runs. The vertical lines indicate time points. Frequency of changepoints between two vertical bars is indicated by shade, the darker shades indicate higher density of change-points.

## 2.4 ESTIMATING UNCERTAINTY IN EXPOSURES

We evaluate uncertainty in the exposure estimates by bootstrapping the mutations. We take the random subset of $N$ mutations by sampling uniformly with replacement from the $N$ mutations in the sample and re-compute exposures trajectories as described above. Bootstrapping mutations allows to obtain the error bar on the exposure trajectories.

## 3 RESULTS

We applied our approach to 2435 whole-genome sequencing samples of 39 cancer types from Pan-cancer Analysis of Whole Genomes (PCAWG) network Gerstung et al. (2017); Dentro et al. (2017). We use 48 signatures provided by PCAWG. Following *Alexandrov et al.*Alexandrov et al. (2013a), we classify mutations into 96 categories based on their three-nucleotide context. A set of mutations is represented as a vector of counts over 96-element vocabulary where $x_k$ is the number of mutations of category $k$ in the set.

### 3.1 SIMULATIONS

To test TrackSig's ability to reconstruct the exposure trajectories, we generate a set of simulated samples with known ground truth. Simulations have 50 time points (average number of time points in PCAWG samples). Each simulation has four active signatures. We generate simulations with 0 to 3 change-points that are placed randomly on the timeline. For each segment on the timeline, we sample signature exposures from a symmetric Dirichlet distribution. Finally, we sample 100 mutations for each time step from the mixture of signature multinomials.

We run TrackSig on the simulated data and compare the reconstructed exposure trajectories to the ground truth. We computed the absolute difference between predicted exposures and the ground truth at each time point and take the median across all time points and all four signatures. On the simulations with no change-points, the median per simulation differences is 1%. On simulations with 1 to 3 change-points, this median increases slightly to 1.9%.

We computed the overall exposure change (maximum exposure minus minimum exposure in the trajectory). The discrepancies between the estimated and ground-truth trajectories is less than 6% in 87% of cases across all signatures in all simulations. To compare the direction of the activity change, we divide signatures into three categories: with decreasing activity, increasing activity and no activity change (if max change is less than 6%). The direction of maximum change is consistent in 95% of all signatures across all simulations.

### 3.2 BOOTSTRAPPING

We assess the variability in exposure trajectories by performing bootstrap on the real data using PCAWG dataset. We sample mutations with replacement from the original set and re-calculate their exposures and change-points. We perform 30 bootstrap runs for each sample. Fig. 1 shows an example of bootstrapped trajectories from leukemia.

Signature trajectories calculated on bootstrap data are stable. The mean standard deviation of exposure values calculated at each time point is 2.7%. We evaluate the consistency of signature changes across the entire exposure trajectory: size of signature change and location of the change-point. The mean standard deviation of the signature change is 4.2% across the bootstraps. This standard deviation does not exceed 5% in 74.8% of samples (does not exceed 10% in 99.5% of samples).

## 4 DISCUSSION

TrackSig reconstructs the cancer evolutionary trajectories through mutation signatures exposures. Our method provides further insight how mutation profile changes throughout tumor development. Previous methods to track changes in the signature exposure have relied on pre-specified groups of mutations, such as those occurring before or after whole genome duplications Alexandrov et al. (2016); Gerstung et al. (2017). In contrast, TrackSig uses the distributions of mutation categories to group mutation, which leads to more sensitive detection of changes in signature exposure.

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
