# OpenReview forum: "Reconstructing evolutionary trajectories of mutations in cancer"
_ICLR.cc/2018/Workshop — Accept_

### Official Review · AnonReviewer1 · 2018-03-07
**Interesting and grounded solution to a complex problem**

**Rating:** 8
**Confidence:** 4

**Review:**

The authors tackles the complex issue of detecting the evolutionary trajectory in cancer by mean of time-evolving mutational signature. The paper is very well written, clear and essential and statistically robust in its methodology. The simulated study provides a reasonably convincing safety check, and the results on the real dataset are indeed interesting. My only concern regards the fact that only a single real dataset has been used: a wider experimental side would definitely provide a stronger support to the paper's claims. Moreover, it would be interesting to check the stability of the method as a function of the authors' choices within the procedure, e.g. the bins' size when partitioning the sorted set of mutations.
Overall, I rate the paper as very interesting and well written, and I recommend it for acceptance.

---

### Official Review · AnonReviewer2 · 2018-03-08
**Novel use of a latent topic model to track mutation signature exposure over time.**

**Rating:** 6
**Confidence:** 4

**Review:**

The authors proposed to use a latent topic model (with fixed and specified topics) to track proportion of each mutation signature (= signature exposure) over time. They propose to use maximum likelihood to identify the change points ( = points in time when these proportions change substantially). They finally show that there is small discrepancy between their predicted mixtures and the ground truth.
The idea sounds interesting. Here are my issues:

- It is not clear how this idea is inherently different from the methods based on mutation clustering, which is used in the past. Also, these traditional methods are not compared against the proposed method.
- The latent model used in the paper (Sec. 2.2) is most interesting when the topics (\mu_i) are unknown. Here the authors assume those patterns are provided a priori (48 signatures in PCAWG). It would be nice to ignore those and estimate them. This gives the method more flexibility (although making it challenging to compare against the ground truth). This anyway could be insightful in identifying new mutation signatures that may not have been found in the past.

---

### Official Review · AnonReviewer3 · 2018-03-11
**Interesting model**

**Rating:** 5
**Confidence:** 3

**Review:**

This submission applies topic models to infer mutational signature exposures (i.e. topic proportions) from cancer mutation data across time. They apply their method to simulations and data from PCAWG. Overall, the model is interesting and potentially useful. I do have some comments on the experiments.  Also, not knowing the literature in cancer genomics deeply, I wonder how this method compares to existing approaches to analyze mutational signatures such as NMF.

1. How accurate are the inferred change points  (using BIC) relative to the true number of change points ?  How accurately can the change points be localized ?

2. While simulating from a symmetric dirichlet, what was the concentration parameter used ? This could affect the difficulty of the inference. For example, if the concentration parameter is small, the signature exposures could be skewed towards zero or 1. It would be good to test their accuracy as a function of the concentration parameter as well as M (the number of components).

3. Was the EM algorithm run with multiple random restarts ?

4. How do the results compare to applying NMF which is the commonly used method  for analyzing these data?

---

### Decision · Program_Chairs · 2018-03-20
**ICLR 2018 Workshop Acceptance Decision**

**Decision:**

Accept

**Comment:**

Congratulations, your paper was accepted to the ICLR workshop.